

# Towards efficient management of riverbank filtration sites: New insights on river–groundwater interactions from environmental tracers and high-resolution monitoring

Krzysztof Janik[1], Arno Rein[2], Sławomir Sitek[1]

[1]Institute of Earth Sciences, Faculty of Natural Sciences, University of Silesia in Katowice, Będzińska Str. 60, 41-200 Sosnowiec, Poland
[2]Chair of Hydrogeology, TUM School of Engineering and Design, Technical University of Munich, Arcisstr. 21, D-80333 Munich, Germany

*Correspondence to*: Krzysztof Janik (krzysztof.janik@us.edu.pl)

**Abstract.** Riverbank filtration (RBF), a managed aquifer recharge (MAR) technique utilised at the river–groundwater interface, can enhance groundwater quantity and quality, thus improving water supply security. However, it demands targeted local and regional monitoring strategies to understand how recharge efficiency and water quality benefits may vary with seasonal and short-term, event-based river flow fluctuations, upstream contaminant inputs, and site-specific aquifer heterogeneity. We evaluated river water–groundwater mixing and groundwater residence times to enhance the knowledge of

aquifer recharge dynamics at the RBF site near Tarnów, Poland, serving as a critical drinking water source for this agglomeration. By coupling environmental tracers (stable water isotopes, chloride concentration, water temperature and specific electrical conductance) with high-resolution hydrological, meteorological and groundwater abstraction records, we show that RBF is the dominant recharge mechanism for the analysed system functioning, constituting over 90% of year-round yield from the production wells near the riverbank. Based on this example, we present a transferable and practical methodology

for managing RBF systems efficiently: a multi-tracer, Ensemble End-Member Mixing Analysis (EEMMA) based workflow that covers at least one hydrological year, checks for local biases, and combines discrete water samples with continuous monitoring of physicochemical and hydrometeorological data, provides a robust and cost-effective template for recharge-source assessment. Such a framework determines both quantitative and qualitative status of abstracted groundwater and facilitates proactive responses to upstream pollution events and/or rapid hydrological shifts, which are crucial for sustainable

water resource management internationally.

## 1 Introduction

Understanding surface water–groundwater interactions is crucial for designing monitoring schemes of hydrogeological systems aimed at protecting drinking water abstraction sites. This can be achieved by providing high-resolution data that can detect contamination early on, thus safeguarding a high-quality water supply and sustaining ecosystem health (Yang et al.,

2011; Lewandowski et al., 2020; Jafari et al., 2021). This is especially relevant to riverbank filtration (RBF) sites, where river water artificially recharges groundwater. RBF, practised in Europe for over a century (Hoang et al., 2024), is now classified as one of the managed aquifer recharge (MAR) techniques (Labelle et al., 2023). The RBF's underlying premise is to induce river water infiltration into the aquifer by pumping groundwater production wells near the riverbank. Consequently, the replenished groundwater can be used, e.g. for drinking or irrigation purposes. During infiltration and flow in the aquifer, the

recharged water (bank filtrate) undergoes natural attenuation processes such as filtration, biodegradation, or adsorption, typically enhancing water quality at relatively low associated costs (Singh et al., 2010; Covatti and Grischek, 2021; Covatti et al., 2023; Labelle et al., 2023; Verlicchi et al., 2024). Thus, in addition to the augmentation of groundwater resources, RBF can improve water quality, which can have particularly strong effects for aquifers of limited thickness and/or lateral extension (Yang et al., 2023).



For groundwater management, it is vital to know subsurface residence times and flow paths of the bank filtrate, as well as the characteristics of mixing with regional groundwater. This also includes groundwater protection and monitoring plans, as well as water supply risk assessment (Bekele et al., 2014; Sitek et al., 2023). Recent studies underscore the value of multiple tracers and high-resolution monitoring for a robust characterisation of surface water–groundwater interaction (Czuppon et al., 2025). Lapworth et al. (2021) combined observations of environmental tracers, groundwater levels and spatially explicit river

discharge rates to assess groundwater–river connectivity and groundwater recharge. Similarly, Jódar et al. (2020) used time series of stable water isotopes, electrical conductivity (EC) and temperature to distinguish fast preferential flow from slow matrix recharge in a limestone aquifer. Duque et al. (2023) comprehensively reviewed methods for studying surface water–groundwater interactions, highlighting the growing interest among researchers in this matter, due to its many implications for integrated water resources management, ecology and contamination assessment. In another study, Jasechko (2019) thoroughly

explored hydrogeological applications of stable water isotopes, pointing out the importance of comprehending groundwater recharge, storage, and discharge.

Many European RBF sites operate in coarse alluvial sediments of similar hydrogeology. Nevertheless, in numerous cases and for various reasons, field investigations remain limited to only basic, low-resolution monitoring, as required by law. Such an approach hinders a complete understanding of how the RBF system operates and responds to changes occurring in the

catchment area due to climate change and human activity. This study hypothesises that river–groundwater interactions at an operating RBF site and the upstream catchment of the river utilised for recharge must be well understood in order to implement and manage such a system effectively. Here, we test such an approach at the example of the Kępa Bogumiłowicka RBF site, located near the city of Tarnów, southern Poland, within the Dunajec River catchment. To achieve the set objective, we carried out detailed, year-long monitoring, including river water and groundwater levels, as well as five environmental tracers in river

water and groundwater: stable water isotopes ($\delta^2$H and $\delta^{18}$O), chloride (Cl$^-$), temperature, and specific electrical conductance (SC). Furthermore, hydrological and meteorological observations at different locations within the Dunajec catchment were used. This allowed evaluating the seasonal isotopic composition of river water and groundwater in the Tarnów region, bank filtrate residence times, and the contribution of river water to the Quaternary aquifer. With the investigated RBF site as an example, our study describes how a comprehensive understanding of hydraulic and geochemical processes, both on the local

and regional scale, is required for sustainable MAR functioning. Based on this example, water monitoring and data analysis recommendations for RBF site operators are presented. For an RBF system, this often requires including the whole catchment of the involved river for monitoring and the interpretation of multiple tracers. Therefore, this study evaluates how upstream hydrological variations can impact downstream local groundwater recharge and abstraction as a function of time. This is also important given climate change and required adaptations, in response to more frequent and more prolonged droughts on the

one hand, and flooding events on the other hand. Despite the clear benefits of including upstream regions in such analyses, catchment-wide monitoring remains relatively uncommon, to date, for investigations and decision-making concerning RBF systems.

## 2 Materials and methods

### 2.1 Study area

The investigated RBF site is located in the village of Kępa Bogumiłowicka near Tarnów, Poland, and run by the public utility company Tarnów Waterworks Ltd. By groundwater abstraction, it provides about 30% of the drinking water supply for the Tarnów agglomeration. The well field lies on the right-bank (eastern) floodplain of the Dunajec, 34 km upstream of its mouth, in the catchment's northern sector (Fig. 1–2). The site hosts 11 production wells, seven of which (S30–S36) are situated about 110 m and four (S37–S40) about 350 m inland of the riverbank (Fig. 1). These vertical production wells are set up as siphon

wells, i.e. they do not have pumps installed and are connected to a collector well (caisson) via a siphon pipe (Bartak



and Grischek, 2018). The capture zone of the well field covers about 4.8 km² (Sitek et al., 2025). The abstracted groundwater is treated with sodium hypochlorite in the collector well and then supplied to the municipal network (Wojtal, 2009). The mean water production capacity reaches up to 10,000 m³ d⁻¹. During the study sampling period (i.e. October 2022–October 2023), the average groundwater abstraction rate (drinking water production) at the well field equalled 9,391 m³ d⁻¹, which was similar
to the mean long-term observation from the period 2015–2025 (9,022 m³ d⁻¹).

Figure 1: Overview of the study area. Average groundwater flow conditions (spring of 2021), groundwater table contour lines and flow directions, as well as the capture zone extension, are based on steady-state groundwater flow modelling results (Sitek et al.,
90   2025).





**Figure 2: The Dunajec catchment within the Polish and Slovak territories, indicating rivers and cities most relevant in this research, as well as considered meteorological and water-gauging stations.**





The unconfined Pleistocene aquifer, composed mainly of alluvial gravel and sand with well-rounded pebbles, overlies thick

Miocene clays and is the area's only aquifer. The RBF site is covered mainly by thin Holocene silts and organic mud; further

inland, clayey silt lenses also occur (Fig. 3).

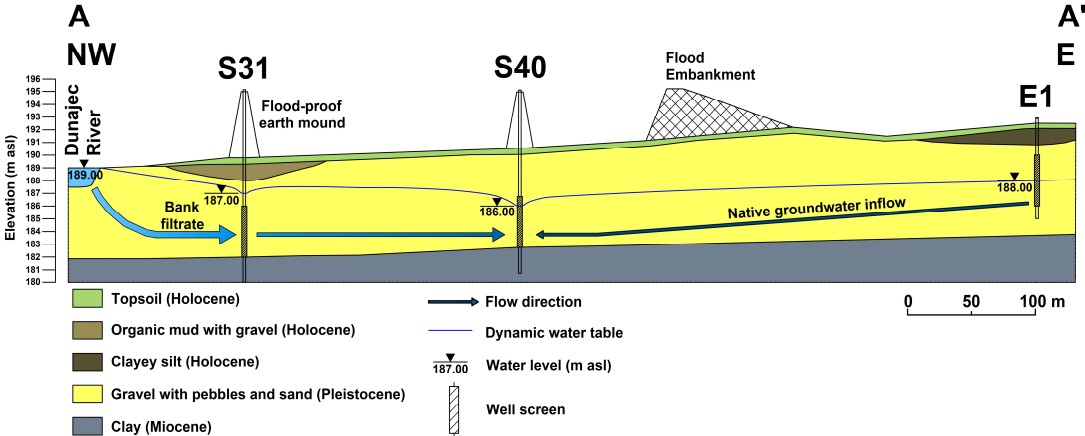

**Figure 3: Schematic hydrogeological cross-section of the study area. The arrows' thicknesses and colours schematically indicate the amount and type of water reaching the production wells. Light blue: influx from the Dunajec, dark blue: native groundwater influx.**
**Thicker arrow stands for higher flux. The flood embankment marks the boundary of the Dunajec floodplain. The cross-section line is marked in Fig. 1.**

The average groundwater level within the site is about 4 m below ground level (bgl), ranging from 2 m bgl in the northern part

to 6 m bgl in the southern part, depending on the Dunajec level and the groundwater abstraction rate. The average aquifer

thickness is 7.5 m, with hydraulic conductivity ($K$) between $4×10^{-4}$ and $3×10^{-3}$ m s$^{-1}$, highest in the northern sector.

Groundwater recharge occurs via infiltration of precipitation and riverbank filtration. Naturally, groundwater flows from the

south-east towards the river, but pumping reverses this flow direction locally, converting the river's character from a gaining

to a losing stream at and near the RBF site (Fig. 1). Basic statistics of selected hydrometeorological parameters from the study

area can be found in Table S1, while meteorological data distribution during the sampling period is shown in Fig. S1.

Hereinafter, all figures and tables with the prefix 'S' refer to the materials presented in the Supplement.

The Dunajec is a 247 km long right-bank tributary of the Vistula River (Fig. 2). The Czarny Dunajec River, its source tributary,

originates in the Western Tatras. In Nowy Targ, it joins with the Biały Dunajec River and forms the main Dunajec River

(Wołosiewicz, 2018; Fig. 2). The Dunajec and its main tributary, the Poprad River, are the only rivers draining the Tatras and

Podhale (Inner Carpathians; Chowaniec, 2009). The Dunajec catchment spans 6,804 km$^2$, with 4,835 km$^2$ in Poland and

1,969 km$^2$ in Slovakia (Kruk et al., 2017). The Dunajec source is 215 km upstream from the studied RBF site location, so about

96% of the catchment area is already drained (Fig. 2).

### 2.2 Analysed datasets

The datasets analysed in this study are characterised in Table S2. R v. 4.4.2 (R Core Team, 2024) was used for the data analysis.

### 2.2.1 Stable water isotopes and chloride concentration - field data

During 12 monthly sampling campaigns conducted from October 2022 to October 2023, 72 water samples were collected at

six sampling points to analyse stable water isotopes ($δ^2$H and $δ^{18}$O) in the study site region (Table 1). The number of samples

was equal for all the locations, i.e. 12 observations per location. Samples were taken from the Dunajec river at the surface

water intake of Tarnów Waterworks (1 km upstream of the well field, Fig. 1) and from five groundwater wells: two production

wells located close to the river in the northern and southern RBF site sectors (S31, S36), two production wells further inland

in the northern and southern RBF site sectors (S37, S39), and the observation well E1 located further east, within the northern





part of the production wells' capture zone (Fig. 1). The Dunajec was sampled at the surface water intake, as access to the river (or more specifically to the continuous flow of the Dunajec), is difficult at the RBF site. This location is also the river water sampling point used by the Tarnów Waterworks. The observation well was used to obtain information on native groundwater, here defined as regional groundwater recharged via precipitation into the Tarnów aquifer system.

**Table 1.** Sampling points for the stable water isotope and chloride analysis (cf. Fig. 1).

| Sampling point | Location | Distance from the riverbank (m) | Well screen depth (m bgl) | Mean drawdown (m) | Mean hydraulic conductivity $K$ of the aquifer (m s$^{-1}$) |
|---|---|---|---|---|---|
| Dunajec River | Surface water intake of Tarnów Waterworks | – | – | – | – |
| Well S31 | RBF site (production wells) | 115 | 3.8–7.8 | 0.35–0.43 | 3.00×10$^{-3}$ |
| Well S36 | | 105 | 5.0–9.0 | 1.40–2.30 | 7.17×10$^{-4}$ |
| Well S37 | | 390 | 3.5–7.5 | 1.35–2.15 | 9.90×10$^{-4}$ |
| Well S39 | | 335 | 3.2–7.2 | 0.48–0.58 | 2.14×10$^{-3}$ |
| Observation well E1 | Capture zone - upstream groundwater flow | 700 | 2.5–6.5 | – | ND (about 5×10$^{-3}$) |

**Mean hydraulic conductivity and drawdown values were obtained from pumping tests (Wojtal, 2009); mean water table drawdowns were calculated at well field production rates of 377.6 m$^3$ h$^{-1}$ and 504.3 m$^3$ h$^{-1}$, respectively; ND – no data.**

River water was sampled about 30 cm below the surface using a 500 ml polypropylene pendulum beaker. Groundwater was collected with a stainless-steel point-source bailer at well screen depth via piezometer tubes installed in each production well.

As the well field operated continuously, the production wells were not additionally pumped prior to sampling. Observation well E1 was pumped before sampling, removing at least three water column volumes from the well. All samples were filtered through a 0.45 µm syringe filter, collected in 2 ml glass vials, and stored at 4–6 °C before shipment to the laboratory. Isotope analyses were performed at Technische Universität Dresden, Germany (Core Facility Environmental Analytics) using high-precision isotope ratio mass spectrometry (IRMS; MAT 253, Thermo Fisher Scientific, Bremen) coupled with a high-

temperature pyrolysis oven HT (HEKATech, Wegberg). Each 1 µL sample was injected six times, with the first measurement routinely discarded. Average analytical uncertainties were 0.14‰ (±0.07‰) for $\delta^{18}$O and 0.66‰ (±0.26‰) for $\delta^2$H isotopes. Laboratory procedures followed the internationally accepted International Atomic Energy Agency (IAEA) standards to ensure analytical quality. Oxygen and hydrogen isotope ratios were expressed in conventional delta ($\delta$) notation, $\delta^{18}$O and $\delta^2$H, as per mille (‰) values relative to the VSMOW2 standard (Sharp, 2017).

Additionally, since January 2023, chloride (Cl$^-$) measurements were carried out monthly at six water sampling locations for stable water isotope analysis (Table 1). In total, from January to October 2023, Cl$^-$ content was measured in situ in 54 samples (nine samples per location, Table S3) using a HACH DR1900 Portable Spectrophotometer (Hach Company, Loveland) with LCK311 Chloride Cuvette Tests.

### 2.2.2 Stable water isotopes and chloride concentration - archival data

Records of $\delta^{18}$O and $\delta^2$H in precipitation were obtained from the Global Network of Isotopes in Precipitation (GNIP) database (Araguás-Araguás et al., 2000) via the IAEA's WISER portal (IAEA/WMO, 2024). The dataset included monthly measurements from the Kraków station (205 m asl), located at the Kraków-Balice international airport (Duliński et al., 2019), approximately 70 km west of Tarnów (Fig. 2). The Kraków station is the nearest GNIP station to the study area with constant precipitation measurements. Precipitation data covers the period 2000–2023 (n = 275 records). A local meteoric water line

(LMWL) was fitted to these observations via ordinary least squares regression (OLSR; Hughes and Crawford, 2012). OLSR was chosen because it provides a long-term, regionally representative isotopic relationship by giving equal weight to all



observations, irrespective of precipitation amount, which avoids seasonal weighting effects and reflects the broader climatic

signal. In Kraków, precipitation is biased toward the summer season, when rainfall is typically more enriched in $\delta^{18}O$ due to

higher temperatures and evaporation (Duliński et al., 2019). As a result, precipitation-weighted mean $\delta^{18}O$ values tend to be

higher than simple arithmetic means. Applying a precipitation-weighted regression in such a context would overemphasise

these summer values, potentially distorting the slope of the LMWL. A water mixing line was also constructed using mean $\delta^{18}O$

and $\delta^2H$ values from the Dunajec and native groundwater (observation well E1). The second-order isotopic parameter

deuterium excess (d-excess or *d*), primarily reflecting the conditions of evaporation at the precipitation moisture source

(Dansgaard, 1964), was calculated for all stable water isotope data (precipitation, river water, and groundwater) using Eq. (1):

$d = \delta^2H - 8 \times \delta^{18}O$                                                                                       (1)

These data were compared to observations of stable water isotopes at other characteristic locations upstream in the Dunajec

catchment. For precipitation, data from three meteorological stations located in the southwest and central south of the

catchment were analysed. These include Liesek (about 8 km west of the catchment boundary, Orava Basin, elevation of 692

m asl; GNIP database), Ornak (Western Tatras, elevation 1100 m asl; Różański and Duliński, 1988) and Stara Lesna (High

Tatras, 721 m asl; GNIP database). For river water, a range of measurements in the Dunajec and the Poprad rivers were

evaluated (Bodiš et al., 2015; Kotowski et al., 2023; see Table S2 for details). For groundwater, numerous observations from

southern Poland were obtained from the Polish Geological Institute – National Research Institute (PGI-NRI). These data were

collected by the PGI-NRI as part of the project "Updating and sharing information resources from the groundwater

environmental tracers database" (PGI-NRI, 2024) and made available to the authors upon request. Only groundwater

observations with a tritium content >0.5 TU were selected from these data for further analysis. This selection ensured that only

young (or recent) groundwater (up to about 60 years old), comparable to groundwater in the Tarnów region, was assessed. In

total, 387 points were considered, 76 of which were within the Polish part of the Dunajec catchment. The spatial distribution

of $\delta^{18}O$ and $\delta^2H$ in young groundwaters across southern Poland was interpolated using the geostatistical kriging method

(Workneh et al., 2024) in ArcMap 10.8.1, ESRI. For Slovakia, only data on precipitation and river water were available.

Archival records from Tarnów Waterworks (September 2022–October 2023) were obtained to enhance our Cl⁻ concentration

measurements in the Dunajec and groundwater. These include 11 records in the Dunajec (sampled at the surface water intake,

Fig. 1), and three records in each of the analysed production wells (S31, S36, S37, and S39).

### 2.2.3 High-resolution monitoring data

Six dataloggers (Levelogger 5 LTC; Solinst Canada Ltd., Georgetown) were installed in the field: one at the surface water

intake in the Dunajec River, four in production wells (S31, S36, S37, and S39), and one in observation well E1 (Fig. 1). These

non-vented devices measure absolute pressure (hydrostatic + atmospheric pressure), temperature (°C), and EC ($\mu S\ cm^{-1}$)

(Solinst Canada Ltd., 2024). Water levels (m asl, relative to a fixed datum at each location) were calculated by barometrically

compensating the absolute pressure data using the Levelogger Software, with atmospheric pressure data from a Barologger 5

device (Solinst Canada Ltd., Georgetown) installed at the RBF site. Manual water level measurements, taken near the time of

a scheduled Levelogger reading start, were used to adjust the compensated water levels to the fixed datum. For further analysis,

raw EC values were converted to specific electrical conductance (SC), corresponding to EC at 25 °C. Also, during each

sampling campaign, manual water level checks were performed using an electric meter (±1 cm accuracy). Measurement

periods differed across the four production wells and the observation well. The longest records were obtained for S36 and S37,

beginning in October 2022; monitoring at S31 and S39 started in November 2022 and at E1 in March 2023. From 6 September

to the end of the observation period (October 2023), river water SC data were unavailable due to a sensor malfunction.

Hydrological and meteorological data were retrieved from the Institute of Meteorology and Water Management – National

Research Institute (IMWM-NRI, 2024). These include weather data and Dunajec flow rates ($m^3\ s^{-1}$) for 2000–2023, measured

at the Zgłobice water-gauging station located approximately 4 km upstream of the studied well field (Fig. 2; daily resolution).



To assess the contribution of upstream Dunajec river flows, observations were also obtained from water-gauging stations
Nowy Sącz, Gołkowice, and Nowy Targ-Kowaniec (Fig. 2; Table S2) for 2000–2022 (IMWM-NRI, 2024). The used
meteorological dataset includes daily air temperature (°C), snow cover (cm), precipitation (mm), and relative air humidity (%)
recorded at three stations: No. 575 – Tarnów (1966–2023), No. 6525 – Polana Chochołowska (2022–2023), and No. 650 –
Kasprowy Wierch (2022–2023; Fig. 2; Table S2). Additionally, hourly data on the drinking water production of the well field
(groundwater water abstraction rate) from October 2022 to October 2023 were obtained from Tarnów Waterworks.

**2.3 Evaluation of river–groundwater mixing at the RBF site**

When selecting a method to assess water mixing, such as in a given production well, it is essential to evaluate if tracer levels
of the recharge sources (end-members) differ significantly from each other and tracer levels of the mixed water (Jasechko,
2019; Kirchner, 2023a). A series of statistical tests was performed to assess these differences. Observations were considered
hydrologically linked (paired): e.g. tracer levels in well S31 were constantly influenced by the river water. The normality of
pairwise differences between sampling locations was tested using the Shapiro–Wilk test (Shapiro and Wilk, 1965). For every
tracer, combinations of the six sampling locations were compared (nine pairs per tracer since there was no need to compare
pairs of production wells). Only field measurements were considered for $Cl^-$ to ensure equal sample sizes across sampling
points. Then, to evaluate if there were statistically significant differences between the sampling locations, a paired t-test (Ross
and Willson, 2017) was used for testing pairs with normally distributed differences, and the Wilcoxon signed-rank test (Bauer,
1972) for testing pairs with non-normal distributions. Source fractions in the abstracted water were estimated for the production
wells S31, S36, S37, and S39 using four environmental tracers: $\delta^{18}O$, $\delta^2H$, $Cl^-$, and SC.

The characteristics of the analysed data precluded the use of common end-member mixing approaches that rely on constantly
distinctive end-member differences (Jasechko, 2019; He et al., 2020), as $\delta^{18}O$ and $\delta^2H$ values in groundwater from the
observation well E1 and river water from the Dunajec were very similar seasonally (for six months). Also, river $\delta^{18}O$ and $\delta^2H$
values and their variation were similar to mixed water from production wells S31 and S36. Therefore, the Ensemble End-
Member Mixing Analysis (EEMMA) was applied to address these limitations. This method uses tracer time series and
reformulates the mixing model as a no-intercept regression (Kirchner, 2023a). Various EEMMA models were applied in R
(using the "EEMMA" package; Kirchner, 2023b), providing summary statistics for end-member contributions. For the
EEMMA modelling, two end-members were defined: the Dunajec and native groundwater. High hydraulic conductivity values
characterise the aquifer, either not isolated or overlain by thin low-permeability sediments (Fig. 3). Due to its limited thickness
and local flow conditions in the Tarnów region, native groundwater is young and isotopically similar to average regional
precipitation (Leśniak and Wilamowski, 2019); thus, precipitation was not treated as a separate end-member. EEMMA models
were first run for each production well without accounting for an unsampled end-member or memory effect, for the measured
stable water isotope and $Cl^-$ concentration time series. EEMMA modelling was then repeated with the memory effect option
enabled, following the method author's recommendation (Kirchner, 2023a). In tracer studies, this "memory effect" refers to
the influence of past mixture compositions on the current state of a mixture. In systems with long residence times, such as
aquifers, part of the observed signal may reflect legacy contributions, complicating interpretation by introducing a temporal
lag between source changes and their detection (Kirchner, 2023a). Due to high measurement frequency, only memory-effect
EEMMA models were applied for SC. Additional EEMMA calculations were performed for $\delta^{18}O$ and $\delta^2H$ values, divided into
three intervals: a period of high mountain meltwater contribution to the Dunajec river water (January–June 2023) and two
baseflow-dominated periods with limited or no snowmelt influence (October–December 2022 and July–October 2023). For
$Cl^-$ concentration, EEMMA modelling was also done for a dataset excluding February, March, and April 2023, when elevated
$Cl^-$ concentrations were observed in native groundwater (observation well E1).



### 2.4 Groundwater residence times

Hourly temperature measured in the Dunajec and in the groundwater of two production wells located close to the riverbank and thus predominantly recharged by the river (S31 and S36) was used as a natural tracer to estimate groundwater residence times (RTs). The analysis relied on identifying the time lag corresponding to the maximum correlation between temperature fluctuations in the river and the wells, assuming that this delay reflects the retarded travel time of inflowing river water, following the approach of Moeck et al. (2017). RTs were evaluated by calculating cross-correlations between river and

groundwater temperature time series in R using the "ccf" function (R Core Team, 2024). The method assumes that temperatures at both locations belong to the same stream tube and that heat transfer properties are uniform along the flow path, with local thermal equilibrium between groundwater and sediment (Hoehn and Cirpka, 2006). Lag times between the Dunajec and each of the two analysed production wells were determined as the time offset from the start of each temperature record corresponding to the peak of the cross-correlation function (Bekele et al., 2014). These lag times were then converted into an

estimate of RTs using a thermal retardation factor ($R_T$), a dimensionless factor that relates solute velocity to thermal front velocity. $R_T$ was calculated using Eq. (2) as the ratio of the volumetric heat capacity of the bulk porous medium to that of water alone (Hoehn and Cirpka, 2006; Bekele et al., 2014; Moeck et al., 2017):

$$R_T = \frac{n_e \rho_w C_w + (1-n_e)\rho_s C_s}{n_e \rho_w C_w}, \tag{2}$$

where $n_e$ is the effective porosity, $\rho_w$ and $C_w$ are the gravimetric density and the specific heat capacity of water, and $\rho_s$ and $C_s$

are the aquifer sediments' gravimetric density and specific heat capacity. In this study, $n_e$ of 0.35, characteristic for the gravel aquifers, was used (Woessner and Poeter, 2020); $\rho_w = 1{,}000 \text{ kg m}^{-3}$ and $C_w = 4{,}186 \text{ J kg}^{-1}\text{ K}^{-1}$ were applied for water parameters, and $\rho_s = 2{,}646 \text{ kg m}^{-3}$ and $C_s = 740 \text{ J kg}^{-1}\text{ K}^{-1}$ were applied for the aquifer mainly composed of quartz (Bekele et al., 2014). This leads to an $R_T$ value of 1.87. The heat flow velocity ($v_T$) was also calculated as the ratio of the linear distance between the riverbank and the given production well to the obtained time shift.

### 3 Results

Sections 3.1 and 3.2 describe isotopic, physicochemical, and hydraulic observations collected for the analysed RBF site, whereas Sect. 3.3 presents quantitative assessments of water mixing and RTs within the aquifer. Finally, Sect. 3.4. broadens the study scope by including isotopic data from precipitation, river water, and groundwater measured across the Dunajec catchment, thus providing a regional perspective for the local findings from the study site. Hereinafter, descriptions regarding

stable water isotopes focus on $\delta^{18}$O, because $\delta^2$H generally follows similar trends. Data on $\delta^2$H are presented to readers throughout the Supplement (Tables S3–S4, and S7–S9; Fig. S2, Fig. S4).

### 3.1 Stable water isotopes

Stable water isotopes were measured in precipitation, river water, and groundwater (Fig. 4c–d, Fig. S2a–b, Table S3). The Kraków station recorded the highest seasonality in monthly precipitation $\delta^{18}$O, with an amplitude of 12.5‰ (Fig. 4c). The

seasonality was also evident in the Dunajec, yet differed from the precipitation pattern. In the river, delta-values were higher (isotopically enriched water) from October to December 2022 and July to October 2023, and lower (isotopically depleted water) during snowmelt-related high-flow conditions in the first half of 2023, especially in February, when the sample was collected during the surge (Fig. 4a and d). Groundwater isotopic signatures in the analysed production wells varied differently, depending on their proximity to the riverbank, with river-proximal wells (S31 and S36) closely resembling the river's isotopic

fluctuations. Wells further inland (S37 and S39) showed less variation throughout the year, more shifted towards native groundwater (well E1), which was enriched the most and fluctuated the least, within approximately 1‰ range (Fig. 4d). Regarding the d-excess, all observation points displayed a similar annual pattern (Fig. S2c). Descriptive statistics for the stable water isotope data are shown in Table S4.





**Figure 4: Mean daily Dunajec River flow rate at the Zglobice water-gauging station (a), daily precipitation in Tarnów in the form of rain (blue) or snow (black) (b), monthly means of δ¹⁸O in precipitation (Kraków station) (c), as well as δ¹⁸O in the Dunajec river water and groundwater in the study area throughout the sampling period (October 2022–October 2023). Red circles on plots (a) and (b) indicate the days of water sampling.**





Figure 5 shows the bivariate relationship between δ²H and δ¹⁸O for all river water and groundwater samples collected in the study area during the sampling period. Observations are plotted against the LMWL and an auxiliary water mixing line. All the

data align close to the LMWL (note the different axes scales). Nevertheless, three main clusters can be distinguished: (i) samples from the river, S31 and S36 occupy the lower-left sector during January–June 2023 and shift toward the plot centre and upper right during October–December 2022 and July–October 2023; (ii) samples from S37 and S39 plot mainly in the centre at intermediate isotope ratios, while (iii) data from the well E1 form a compact group that coincides with the long-term (2000–2023) mean from Kraków precipitation. Together, the data points outline a continuum from the most depleted (for half

a year) river and river-proximal groundwater to progressively more enriched inland (river-distal) and background (native) groundwater. For regional context, Fig. 5 includes the median δ¹⁸O (-10.9‰ ±1 standard deviation) of 76 groundwater samples collected within the Dunajec catchment (PGI-NRI, 2024).

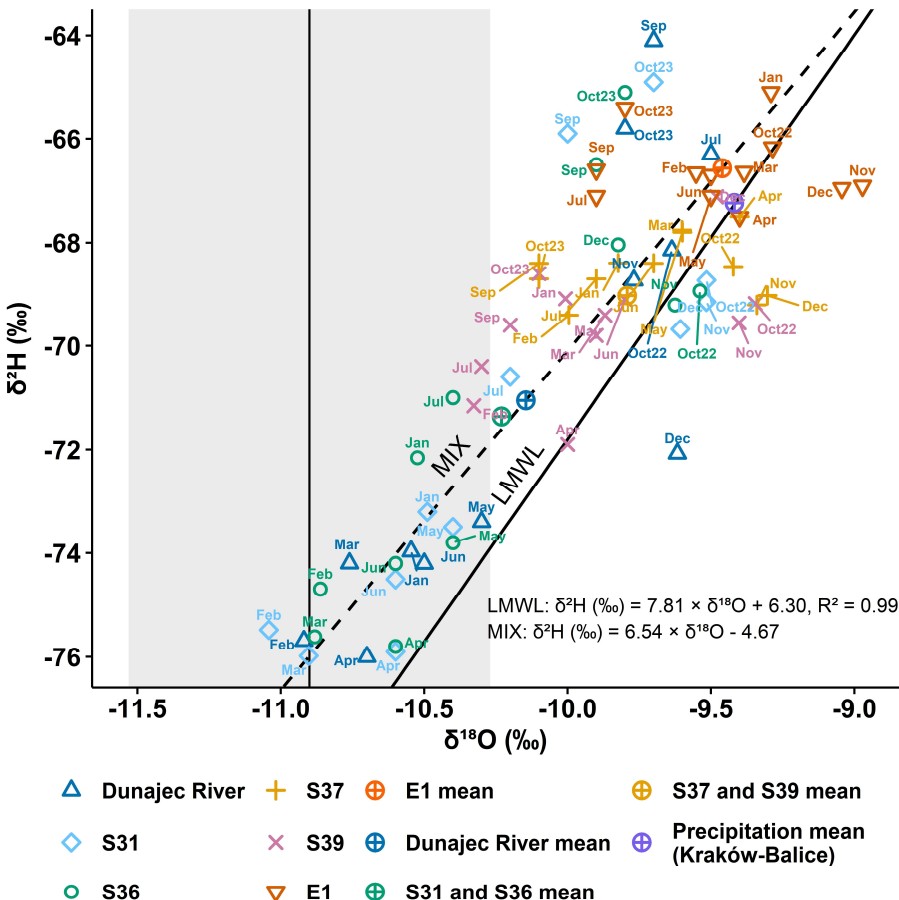

**Figure 5: Dual-isotope plot (δ²H vs. δ¹⁸O) showing delta-values of precipitation (Kraków), Dunajec river water, and groundwater samples from observation (E1) and production (S31, S36, S37, S39) wells. LMWL is the local meteoric water line, MIX is the water mixing line. Colours differentiate sampling locations, with symbols indicating monthly values and mean compositions. The vertical black line is the median value of δ¹⁸O in young groundwater within the Dunajec catchment. The grey area is the median ±standard deviation.**




**3.2 Water levels and physicochemical parameters**

Specifying the differences between values of individual parameters at the analysed sampling points, especially between the end-members, is required for further inference about, for example, water mixing ratios. Also, high-resolution data enables tracking the aquifer's response to rapid river flow changes or halting groundwater abstraction. Thus, this section summarises

the seasonal variation of water levels, temperature and SC in the Dunajec, four production wells (S31, S36, S37, S39) and the observation well E1, registered with dataloggers (Fig. 6a–c, Fig. S3a–c; cf. Methods Sect.), together with $Cl^-$ concentrations determined from discrete samples (Fig. S2d, Fig. S3d; Table S3). Mean daily groundwater abstraction rates (in $m^3\ h^{-1}$) of the well field are included to identify operational stoppages during the sampling period (Fig. 6d). All descriptive statistics are shown in Tables S4–S5, whereas archival $Cl^-$ concentrations in the Dunajec and groundwater from the production wells are

presented in Table S6.

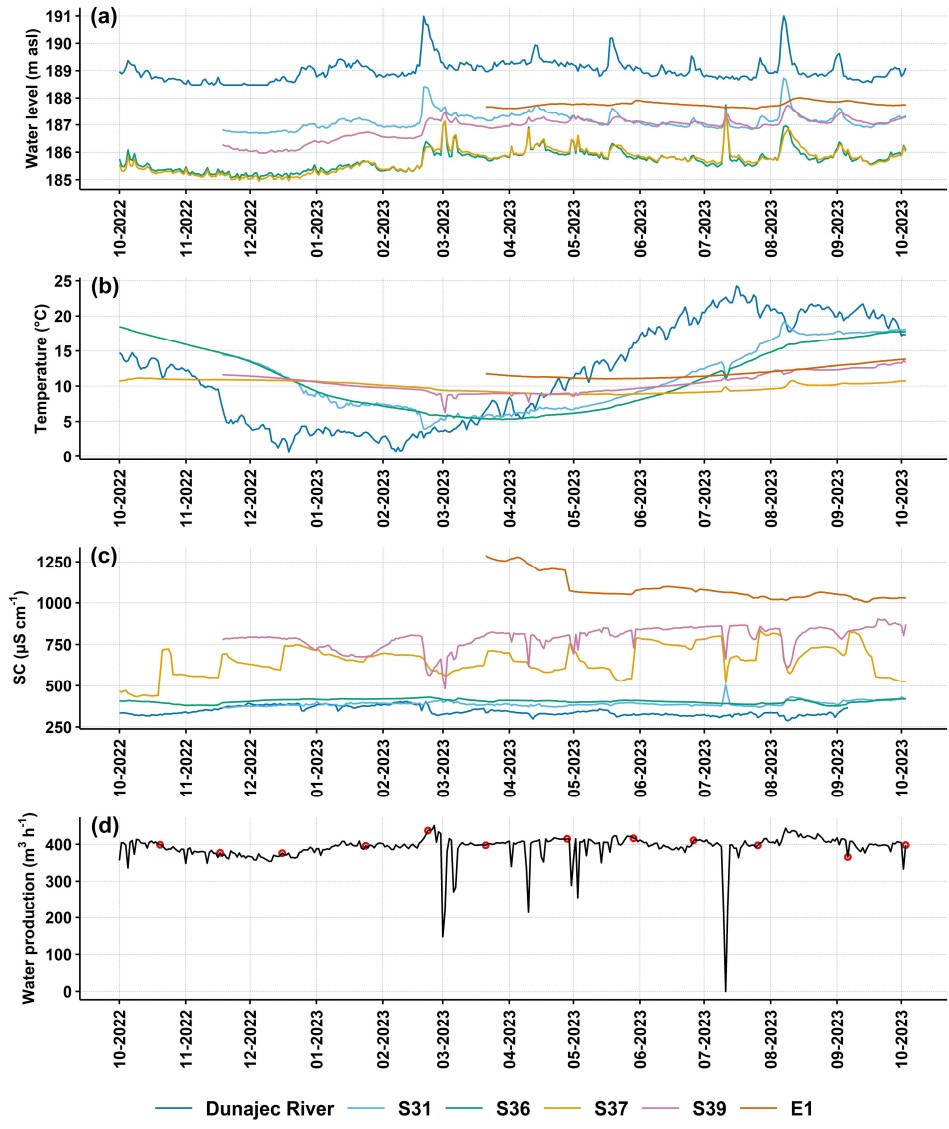

**Figure 6: Datalogger (sensor) observations of mean daily Dunajec and groundwater levels (a), temperature (b), and specific conductance - SC (c), as well as the mean daily water production (groundwater abstraction rate) of the analysed well field (d). Sampling days are marked as red circles on (d).**



During the analysed period, seasonality was apparent in the river data: after a low-flow period in November and the first half of December 2022, the river level remained consistently elevated (with short periods of lower flows, e.g. in the first half of July 2023) as a result of either snowmelt or rainfall across the catchment (cf. Fig. S1, Fig. S5–S6).

Groundwater levels in the production wells followed the river pattern, but with site-specific amplitudes. In general, levels were higher in the northern wells (S31 and S39; Fig. 6a), due to higher hydraulic conductivity (Table 1). The levels in river-proximal

wells S31 and S36 fluctuated by about 2.2 m and 3.2 m, respectively, with their maxima coinciding with each river peak or water abstraction stoppage (e.g. in July, Fig. 6d). One of the river-distal production wells, S37, exhibited a trend almost identical to that of S36, whereas groundwater level responses in S39 displayed a marginal lag to the Dunajec rises. The lowest groundwater level fluctuations were noted in the observation well E1 with an amplitude of about 0.5 m (Fig. 6a), showing a reaction to the most intense precipitation events.

River water demonstrated the highest variability in temperature among observation points, averaging 11.6 °C, with a maximum in July and a minimum in February (Fig. 6b). Mean SC of the Dunajec was 345 µS cm$^{-1}$, with the highest values observed in February, just before a major meltwater surge, which then notably lowered the SC (Fig. 6c). Temperature and SC records in the production wells followed similar trends to those observed for stable water isotopes, i.e. S31 and S36 exhibited values more closely aligned with those of the Dunajec, whereas S37 and S39 showed greater similarity to native groundwater (Fig. 6b–

c). However, the SC of groundwater in S39 resembled SC recorded in E1 more, while S37 appeared more similar to river water, contrary to stable water isotope results. Still, these values were about twice as high as in the river (Table S5), and throughout the whole period, they remained roughly in the middle between the E1 well and the river (Fig. 6c). Rapid SC changes in the production wells were also noted, similar to groundwater levels, in response to river level peaks or temporary shutdowns of well field operations (Fig. 6c–d). Additionally, Figures S7–S10 show hourly records of water levels and SC in

the Dunajec and four analysed production wells, along with the hourly groundwater abstraction rate at the RBF site.

Cl$^-$ concentrations in the Dunajec remained within a narrow range during the sampling period, with S31 and S36 closely following the river values; slightly higher values were recorded in river-distal production wells (Fig. S2d, Tables S3–S4). Native groundwater showed substantially higher and more variable Cl$^-$ concentrations, ranging from 63.0 mg L$^{-1}$ in January to 115.0 mg L$^{-1}$ in March 2023 (Fig. S2d; Tables S3–S4).

**3.3 Water mixing and RTs within the RBF site**

Statistical tests were performed to select a proper method for assessing the degree of water mixing in the production wells, and their results are presented in Table S7. No significant differences were found between nine pairs of observations; therefore, it was decided to discard the conventional mixing approach and use the EEMMA models (cf. Methods Sect.). The models were applied to time series of four tracers (δ$^{18}$O, δ$^2$H, Cl$^-$ and SC). River water fractions in the groundwater of the production wells,

considering all used tracers, were 92–100% for S31, 87–100% for S36, 39–51% for S37, and 46–59% for S39 (mean values ± standard error of the estimated mixing fraction (SE); cf. Table S8). Thus, farther inland from the river, native groundwater constituted a notably higher fraction of the abstracted groundwater.

Fraction estimates for the meltwater contribution period (January–June 2023) closely matched those for the whole observation period (Table S8). Conversely, for the periods with no upstream snowmelt signal (October–December 2022 and July–October

2023), most models yielded statistically non-significant end-member fractions (p > 0.05), with markedly higher SE (Table S9). Temperature time series of the Dunajec water and groundwater in the production wells (S31, S36, S37, and S39) and the observation well E1 are shown in Fig. 6b (Sect. 3.2). Considering stable water isotope results and datalogger records, continuous inflow of native groundwater to S37 and S39 can be assumed, attenuating thermal signals originating from the Dunajec. Therefore, RTs were calculated for S31 and S36 only. The estimated lag times were about 34 days for S31 and 41

days for S36, and after taking $R_T$ (1.87) into account (cf. Methods Sect.), estimated RTs equalled 18 and 22 days, for S31 and S36, respectively (Table 2).





**Table 2.** Estimated groundwater residence times (RTs) between the Dunajec and production wells S31 and S36.

|  | Unit | S31 | S36 |
|---|---|---|---|
| Distance from the riverbank | m | 115 | 105 |
| Estimated lag times | hours (days) | 807 (34) | 987 (41) |
| Estimated RTs (lag time/$R_T$) | hours (days) | 432 (18) | 528 (22) |
| $v_T$ without $R_T$ | m h$^{-1}$ (m d$^{-1}$) | 0.14 (3.42) | 0.11 (2.55) |
| $v_T$ with $R_T$ | m h$^{-1}$ (m d$^{-1}$) | 0.27 (6.39) | 0.20 (4.77) |

**$R_T$ – thermal retardation factor (here: 1.87); $v_T$ – heat flow velocity**

### 3.4 Observations in the Dunajec catchment

Including catchment-wide isotope data allows for evaluating whether the river water recharging the aquifer is isotopically representative of the river's larger source area and/or regional precipitation. This broader perspective is essential for judging how observations downstream may transfer to upstream RBF sites under changing upstream inputs and climate conditions. Thus, this section presents stable water isotopes of precipitation, river water and groundwater, observed across the Dunajec catchment (Fig. 7, cf. Table S2 for observation periods). Also, for a wider context, Fig. 7 locates the site within the southern Poland isoscape (PGI-NRI, 2024; cf. Methods Sect.).

Precipitation, observed at the three meteorological stations in the southern Dunajec catchment (Liesek, Ornak, and Stara Lesna), was, on average, isotopically more depleted than at the GNIP station Kraków, closest to the study site (Fig. 7). River water was likewise slightly more depleted upstream than downstream the catchment. In the region of the analysed RBF site, Dunajec was thus more depleted than precipitation and native groundwater and partly closer isotopically to upstream river water. This pattern was especially evident during a high mountain snowmelt contribution (January–June 2023), when the Dunajec in the study area had a mean $\delta^{18}$O value of -10.62‰.

The spatial distribution of $\delta^{18}$O in young groundwater of southern Poland tends towards higher delta-values in the west and at lower elevations. In comparison, more depleted groundwater occurred in the east and at higher altitudes (Fig. 7). Within the Polish part of the Dunajec catchment (where data were available), enriched groundwater prevailed in the northern areas. Further south, as elevation increased, isotopic depletion was more pronounced, where the Tatras represent the lowest $\delta^{18}$O values across southern Poland (Fig. 7).





**Figure 7:** Distribution of δ¹⁸O in precipitation, river water, and young groundwater across the Dunajec catchment on the Polish and
Slovak territory (constituted by the Poprad catchment). The distribution of δ¹⁸O in young groundwaters in southern Poland is also
shown for a broader perspective. White isolines show the mean annual precipitation (in mm) in the Polish part of the Dunajec
catchment from 1981–2010, based on Kruk et al. (2017). "% of discharge" refers to the mean Dunajec flow rate in Tarnów calculated
for 2000–2022 (water-gauging station: Zgłobice). Thus, e.g. 87% in a water-gauging station upstream indicates that 87% of the
Dunajec flow rate noted at Zgłobice was recorded on average from 2000–2022 at this station. Digital Elevation Model (DEM) source:
NASA Shuttle Radar Topography Mission (2013).



## 4 Discussion

This study shows that RBF is the dominant recharge mechanism for the analysed well field functioning, especially the seven production wells in an array closer to the riverbank (Fig. 1), revealing a strong spatial gradient in aquifer recharge dynamics. The results extend beyond a single catchment and contribute to best MAR practices internationally by providing a quantitative benchmark for assessing vulnerability and optimising RBF schemes across different regions. The following sections discuss our observations in a catchment context and examine their implications for water resources management and monitoring.

### 4.1 Seasonal patterns of river recharge traced by isotopic signals

Seasonal recharge patterns driven by snowmelt and rainfall were reflected in tracer data, supporting similar conclusions by López-Moreno et al. (2023). The substantial contribution of snowpack-derived water to river flow suggests that snowmelt timing and intensity changes could directly affect groundwater recharge dynamics. Our findings highlight the critical influence of the seasonal Dunajec regime on the isotopic composition and recharge dynamics of groundwater abstracted at the studied RBF site. Snowmelt in the Tatra Mountains (Fig. 2; Fig. S5–S6) drove much of the seasonality of the isotopic signal in the river water. The depleted isotopic signatures detected in groundwater during high-flow conditions from January to June 2023 directly reflected meltwater contributions from the upper catchment. Such contributions constitute an essential recharge source, particularly evident in wells nearest the riverbank (S31 and S36). Conversely, during low-flow periods when river discharge is primarily sustained by regional baseflow and/or rainfall (e.g. July to October 2023), isotopic signals were enriched, resembling native groundwater (here observed in the well E1). Thus, the isotopic composition of groundwater abstracted by the river-proximal wells clearly mirrors the dominant hydrological processes occurring upstream in the catchment. For a more thorough temporal interpretation of the Dunajec regime and its influence on abstracted groundwater isotopic composition, we refer to Sect. S1, where we divide the observation period into three time intervals during which different sources constituted the river flow, and interpret each interval in more detail.

From a practical management perspective, the observed isotopic variations are powerful indicators for tracing recharge sources and identifying periods when the aquifer is most vulnerable to contamination from upstream sources. Therefore, knowledge of these seasonal isotopic dynamics enables targeted, event-driven monitoring rather than merely routine observations several times a year (regarding raw water in the production wells). Moreover, maintaining the balance between abstraction and recharge will be increasingly important in regions where climate-driven variability affects surface water availability and aquifer resilience. As climate change increasingly alters snowmelt timing and intensity across mid-latitude regions (Gottlieb and Mankin, 2024), understanding these patterns becomes crucial for sustainably managing water resources at similar RBF sites internationally.

### 4.2 Methodological insights into recharge-source identification

Xia et al. (2020) point out that accurate partitioning of recharge sources is essential for aquifer management and pollutant source tracking. However, overlapping end-member signatures and seasonally transient mixing often hamper it. We addressed this challenge with the recently proposed Ensemble End-Member Mixing Analysis (EEMMA; Kirchner, 2023a–b) applied to four tracers ($\delta^{18}$O, $\delta^2$H, Cl⁻, and SC). The EEMMA models effectively quantified the fractions of river water and native groundwater in abstracted groundwater, especially during periods influenced by upper catchment snowmelt. Our results demonstrate that the transport of bank-filtrated river water to the nearest production wells (about 100 m from the riverbank) constitutes over 90% of their yield year-round, underscoring the vulnerability of the aquifer to river contamination events potentially occurring upstream. The wells farther inland (about 300–400 m from the riverbank) show more attenuated seasonality and abstract a river water–native groundwater mixture influenced more by regional groundwater inflow from the land side. Still, the contribution of river water to these wells remains highly relevant, accounting for approximately 40–60% (Table S8). Notably, the predominance of statistically non-significant results was present during non-meltwater periods (Table





S9). In such hydrological conditions, when the river highly resembles native groundwater due to dominant baseflow or rainfall contribution only, determining individual end-member fractions becomes impractical. However, when these months were part of a broader dataset, including periods with stronger isotopic contrasts, EEMMA models still produced reliable results
comparable to those for meltwater-dominated months alone (Table S8). This indicates upstream control of the river water–groundwater mixing calculations downstream and suggests that limited sampling or datasets restricted to baseflow-dominant periods may lead to high standard errors and non-significant outcomes in the end-member mixing analysis.

We also learned that using a single tracer can be insufficient at some RBF sites and lead to misinterpretation of the water mixing process. At our test site, Cl⁻ proved sensitive to potential local anthropogenic inputs and increased the apparent river
fraction in the wells further inland (Table S8). When the three months with the highest Cl⁻ concentrations recorded in E1 were excluded, the river fraction in S37 and S39 dropped by about 10%, confirming the bias (Table S8). Cross-checking with stable water isotopes and SC prevented overestimation and provided a more reliable, averaged picture of the river water contribution to the river-distal wells (cf. Results Sect.). Similar scrutiny is advisable wherever urbanisation, agriculture or industrial pollution may affect conservative ions. Our study additionally revealed a memory effect in tracer signals, particularly during
hydrological transition periods (e.g. from a snowmelt to a rainfall-dominated regime). Such effects highlight potential pitfalls when relying solely on short-term or low-resolution groundwater sampling, as aquifer retention processes might delay or mask actual recharge dynamics. Thus, reliable attribution of recharge sources requires either comprehensive datasets covering various hydrological scenarios or the integration of various tracers resilient to ambiguous signals with high-resolution datalogger records (see the next Sect.). Extension of this inference about the contribution of river water and groundwater to
production wells' recharge can be found in the Supplement (Sect. S2).

**4.3 Aquifer response to river stage and RBF site operation changes**

High-frequency monitoring is indispensable at RBF sites: any rapid change in river stage and/or deterioration in river water quality can rapidly propagate to production wells if the RTs of bank filtrate in the aquifer are short. Low-resolution sampling, therefore, risks missing the entire range of flow regimes (Gentile et al., 2024) or operationally critical events such as flood-
wave breakthroughs. High-resolution continuous monitoring of physicochemical parameters such as temperature or EC enables the identification of short-term pollution pulses, such as industrial spills, that could otherwise go undetected in low-frequency manual sampling regimes (Coraggio et al., 2022). It also may help distinguish natural seasonal signals, like snowmelt or storm runoff, from anthropogenic impacts, improving response strategies. Intensifying extreme weather events such as floods and droughts further amplify this vulnerability, making continuous, multi-parameter time series essential for
detecting and managing fast aquifer responses that would otherwise go unnoticed (Corona et al., 2023). Popp et al. (2021) confirm this need and show that high-resolution data are vital for quantifying mixing and travel times at the river–aquifer interface.

By integrating high-resolution river and groundwater data with multiple tracers, our study provides a concrete example illustrating how these recommended monitoring strategies translate into actionable insight. The presented setting enabled
observation of the aquifer's rapid response to river flow and groundwater abstraction changes. Increases in the Dunajec level, caused by snowmelt or heavy rainfall, were reflected in all production wells within hours (Fig. S7–S10), indicating strong hydraulic connectivity, high riverbed permeability, as well as high aquifer diffusivity (Welch et al., 2013). This highlights the need to manage river water pollution sources effectively, as contaminants can quickly spread through aquifers with high permeability, which is increasingly critical, given the negative impacts of climate change on surface water quality (Li et al.,
2024). It must also be noted that the impact of extreme Dunajec flows can only be mitigated to a limited extent by the retention reservoirs located 33 and 45 km upstream, respectively, since they have lost most of their retention capacity due to sediment accumulation, and their ability to control floods has diminished (Absalon et al., 2023; Pieron et al., 2024). Groundwater levels also reacted instantly to changes in groundwater abstraction rate: the wells S36 and S37 displayed the largest drawdown



rebounds when pumping was paused, underscoring their sensitivity to operational decisions. For SC, greater variability was observed among the monitored wells. Higher Dunajec levels consistently resulted in lower river water SC and also reduced SC in S37 and S39 (Fig. S7–S10). This reflects increased river inflow reaching more distant wells during high and extreme flow conditions. Conversely, at the peak stage, SC increased in the bank-proximal wells S31 and S36, plausibly because higher river pressure suppressed natural discharge nearby and diverted more mineralised native groundwater towards these cones of depression. Groundwater abstraction shutdowns resulted in a decrease in SC in wells S37 and S39, likely due to the faster inflow of low-SC river water from the west compared to the slower native groundwater inflow from the east, where aquifer transmissivity is lower. Short-term system operation stoppages caused little or no noticeable SC change in wells closer to the river. One exception worth pointing out was S31 in July 2023, when a two-day maintenance pause in well field operation led to an SC increase within an hour, likely due to a rapid inflow of native groundwater (Fig. S7b–c). It is also possible that water from nearby wells S30 and S32 (Fig. 1), where SC is typically higher (Janik et al., 2024), contributed to this increase. After restarting the system, SC in S31 returned to its previous levels within an hour (Fig. S7b–c).

Given the growing affordability of such monitoring solutions and the wide availability of equipment such as dataloggers that continuously record high-resolution data (e.g. water level fluctuations, temperature and EC but also other parameters like nitrogen compounds or Cl⁻), it may radically change the ability to observe the impact of short-term events and the sensitivity of RBF systems to such events. Combining a properly designed monitoring network with constant data transmission to an online server may constitute a new, better approach to designing early warning systems.

### 4.4 RTs estimation using heat tracer

Hourly temperature data from the Dunajec, S31, and S36 were used to estimate the bank filtrate's RTs and heat flow velocity ($v_T$). Heat is a cost-effective, natural alternative to dye tracer experiments, where the recovery rate is often low (Healy, 2010; Moeck et al., 2017) and is increasingly being used in groundwater studies (Kurylyk et al., 2017; Ren et al., 2018), also including MAR systems (des Tombe et al., 2018; Caligaris et al., 2022). The primary goal of implementing this approach was to quantify how quickly river water reaches the production wells near the riverbank. Such knowledge allows for the optimisation of RBF site operation (e.g. in designing sustainable pumping regimes or ensuring that recharge rates align with abstraction rates without overexploiting the aquifer) but also aids in assessing groundwater vulnerability to contamination, as short RTs imply strong hydraulic connectivity between the river and the aquifer, and thus short pollution travel times. Also, short RTs may adversely affect RBF systems since the time needed for natural attenuation and purification processes during infiltration, which may remove or reduce pathogenic microbes and various inorganic and organic pollutants, may not be sufficient (Sitek et al., 2025).

RTs of 18 and 22 days from the river to the wells S31 and S36, respectively, confirmed the favourable hydrogeological properties of the analysed aquifer in both northern and southern parts of the RBF site. While this supports efficient site operations, it also highlights that the seven production wells closer to the riverbank (Fig. 1) may be more vulnerable to potential upstream river water contamination. Given the strong hydraulic connection between the Dunajec and the wells, high-resolution monitoring of abstracted groundwater and river water is essential for early detection of transient contamination events and proactive risk management.

We must also acknowledge the limitations of temperature time series analysis, as noted by Moeck et al. (2017). These include the need for a sufficiently large time offset between locations, the assumption that temperature signals originate from the same stream tube, and uniform heat transfer properties along the flow path. The cross-correlation method assumes linearity and stationarity, yet real groundwater flow is subject to heterogeneity and transient conditions, which may affect the accuracy of travel time estimates. Also, changing the assumed effective porosity value can substantially alter the value of $R_t$ and, consequently, lengthen or shorten the estimated RTs, as Bekele et al. (2014) pointed out.



### 4.5 Recommendations for efficient RBF sites management

This research provides new insights into river–groundwater interactions at an RBF site characterised by a highly permeable aquifer, short RTs, and strong hydraulic connectivity with the adjacent river. Table 3 summarises the key methodological lessons from this study, transferable to other RBF sites: it lists each monitoring or analysis element, explains the added value it can bring to RBF assessments, notes what can be missed without it, and provides associated, indicative net costs.

**Table 3. Methodological takeaways for RBF site operators.**

| Methodological element | Added value for RBF site operation | Potentially missed or misinterpreted aspects without the methodological element | Estimated cost (net price, EUR – July 2025) |
|---|---|---|---|
| **High-resolution (e.g. hourly or sub-hourly) river and well logging (water level + temperature + electrical conductivity or specific electrical conductance), preferably with online access through a dedicated dashboard/platform** | • Detects breakthrough of flood waves or contaminant pulses within hours<br>• Quantifies aquifer diffusivity<br>• Estimates RBF effectiveness<br>• Increases knowledge of the RBF system and groundwater residence times (RTs)<br>• Enables better management of individual wells (e.g. immediate notification of level drops or conductivity spikes enables rapid shutdown) | • Transient and/or quick-time events remain invisible (e.g. delay in response; higher risk of contaminated water entering the water supply system)<br>• RTs hard to estimate | • Telemetry logger for sending data to the server: €1500<br>• Datalogger for measuring absolute pressure, temperature, and electrical conductivity: €1700<br>• Alarm functionality (SMS/email/FTP), e.g. for low water level or conductivity spike: €145<br>• External antenna: €65<br>• Vented cable (probe–logger connection, price per metre): €10<br>• Interface cable (logger–computer connection): €140<br>• Dedicated online platform for data visualisation and analysis: €1440<br>• Software: €260<br>• Installation, programming and delivery: €200 |
| **Multi-tracer design (e.g. $\delta^{18}$O, $\delta^2$H, Cl⁻, EC)** | • Cross-checks end-member fractions<br>• Compensates for tracer-specific biases | • Single-tracer studies may over- or under-estimate the river water fraction in the production wells | • $\delta^{18}$O, $\delta^2$H: € 60 per sample<br>• Cl⁻: HACH Chloride cuvette test 1-70 mg/L / 70-1000 mg/L Cl⁻, 24 tests: €120<br>• EC measured with dataloggers |
| **Coupling discrete samples with continuous loggers** | • Links chemical shifts to hydraulic triggers in real time<br>• Supports event-based management | • The source of sudden quality changes remains speculative<br>• Delayed operational response | • Does not apply |
| **EEMMA mixing analysis** | • Separates overlapping end-members<br>• Identifies lagged signals due to aquifer storage (memory effect) | • Classical two-component models fail when isotope signatures converge | • Free of charge: modelling is performed in the freeware R environment (e.g. in RStudio). |
| **Sampling across the full hydrological year (e.g. monthly)** | • Captures high- and low-contrast periods,<br>• Stabilises statistics | • Datasets limited to low-flow months may yield high SE<br>• Many results become statistically non-significant | • Cost of travel to the site and bottles for sampling, presumably several dozen euros per sampling campaign |
| **Catchment-scale context monitoring** | • Explains seasonality in tracer signals<br>• Improves vulnerability assessment to upstream pollution | • Local monitoring alone misattributes changes to site-scale factors<br>• Early-warning potential lost | • Difficult to estimate; hydrological and meteorological data are often available for free online (or from other well fields upstream based on knowledge/data exchange) |
| **Site-specific numerical model (e.g. MODFLOW / FEFLOW)** | • Tests pumping scenarios<br>• Predicts capture zones<br>• Optimises monitoring well placement | • Over- or under-estimation of groundwater capture zone extent<br>• Less targeted sampling<br>• Location of new production wells and/or observation wells harder to evaluate | • Software licence (if commercial) + analyst time; open-source options free of charge |



**5 Conclusions**

This study provides practical insights for managing RBF systems exploiting shallow, highly permeable aquifers. A multi-tracer, EEMMA-based workflow that (i) covers at least one full hydrological year, (ii) vets each tracer for local biases, and (iii) combines discrete samples with continuous physicochemical parameters records (ideally with real-time online access), provides a robust and cost-effective template for recharge-source assessment at other RBF sites. Such a framework improves understanding of the processes governing water quality and quantity, underpins more reliable vulnerability assessments and

faster operational responses to transient contamination and/or rapid hydrological shifts. Tailored, site-specific strategies can significantly enhance data reliability, which supports real-time management decisions and strengthens drinking water systems' resilience under growing hydrological uncertainty. Therefore, we can assume that investing in high-resolution monitoring networks is justified, especially as climate change intensifies extreme events that disrupt seasonal hydrological patterns (Meresa et al., 2022). Our study also demonstrates that interpreting RBF site data, especially in river-downstream well fields,

requires a catchment-scale perspective for proper inference regarding, e.g., groundwater recharge sources. However, alternative strategies are available for waterworks managers who cannot invest time and/or money in broader catchment monitoring. One option is discharge-dependent sampling at the local scale, with higher sampling frequency during hydrologically sensitive periods such as snowmelt- or rainfall-driven surges. This approach improves the ability to detect short-term variations, enhancing data reliability. Collaboration with well fields located in the upstream parts of the catchment may

also be a cost-effective alternative, allowing data exchange when high-resolution monitoring equipment is unavailable. This study also shows that, for isotope-based assessments, a full hydrological year is a minimum to characterise an RBF system performance adequately. Where isotope sampling is limited in frequency, it should be supplemented with high-resolution data on basic physicochemical parameters, hydrological conditions, and weather.

**Code and Data Availability**

Pre-processed data regarding stable water isotopes, chloride, datalogger records, water production, and hydrometeorological observations, along with the R scripts for the data analysis and visualisation, are publicly accessible on Zenodo (Janik, 2025). Archival data on stable water isotopes in young Quaternary groundwaters in southern Poland were obtained on request from the PGI-NRI. Archival data on stable water isotopes in the Poprad and Dunajec rivers were obtained from the literature and on request from the State Geological Institute of Dionýz Štúr. Hydrometeorological data are also available from the website

https://danepubliczne.imgw.pl/ (in Polish). The Authors recommend using the "climate" R package (Czernecki et al., 2020) to download the IMWM-NRI data.

**Author contribution**

**AR:** Conceptualization, Formal analysis, Validation, Writing - review & editing. **KJ:** Conceptualization, Resources, Data curation, Investigation, Methodology, Visualisation, Formal analysis, Writing - original draft, Writing - review & editing. **SS:**

Conceptualization, Methodology, Supervision, Formal analysis, Funding acquisition, Validation, Writing - original draft, Writing - review & editing.

**Competing interests**

The authors declare that they have no conflict of interest.



**Disclaimer**

We declare that, during the preparation of this work, we used generative AI exclusively for language editing purposes. The content was thoroughly reviewed, and we take full responsibility for the quality of this publication.

**Acknowledgements**

The authors would like to thank Tadeusz Rzepecki, the former President of Tarnów Waterworks, and the Tarnów Waterworks employees (Barbara Jędrzejowska, Tomasz Włodarczyk, Grzegorz Wojtal, and Katarzyna Szczepanek) for allowing us to
conduct research at the Kępa Bogumiłowicka RBF site and providing all the necessary information. We would also like to thank Dr.-Ing. Diana Burghardt from the Core Facility Environmental Analytics (CFEA) of TU Dresden for performing stable isotope analyses, as well as Dr Dana Vrablikova from the Water Research Institute and RNDr. CSc. Juraj Michalko, a researcher at the State Geological Institute of Dionyz Stur, for their assistance in obtaining stable water isotope data from the Poprad and Dunajec rivers.

**Financial support**

The article was co-funded by The Institute of Earth Sciences, University of Silesia in Katowice, the Doctoral School at the University of Silesia in Katowice, and by the project: „jUŚt transition – Potencjał Uniwersytetu Śląskiego podstawą Sprawiedliwej Transformacji regionu" (FESL.10.25-IZ.01.0369/23-003). The project is implemented under the European Funds for Silesia 2021-2027. Program co-financed by the Just Transition Fund.

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
