# Peer review of "Contents of the Supplement"

_EGUsphere, 2025_

## Author Comment (AC1)

Sosnowiec, 2 October 2025

**Authors' response to Referee #1**

**RC1: 'Comment on egusphere-2025-3841', Anonymous Referee #1, 29 Sep 2025**

**General comment**

It was a pleasure to read and review this well-prepared manuscript by Janik et al.. The objectives, methods, presentation and clarity of writing are first class. The manuscript is well focussed, direct and purposeful and will be a significant contribution to RBF/MAR. It makes a valuable contribution that will be well recieved by water resources managers, especially as it gives a great deal of design and monitoring information that can be readily adapted to proposed RBF/MAR systems. I recommend that themanuscript be accepted more or less as s, subject to a few minor editorial suggestions in the attached annotated pdf file. Congratulations on an excellent paper.

REPLY: We sincerely thank you for carefully reading our manuscript and for the very positive and encouraging words. We greatly appreciate your recognition of the clarity, focus, and potential contribution of our work to the field of RBF/MAR. We are particularly grateful for your note on the practical value of the design and monitoring information, as our intention was precisely to provide material that water resources managers and practitioners can directly adapt. We also acknowledge your editorial suggestions provided in the annotated PDF. We will implement these minor revisions to improve the quality of the manuscript. Please find below our answers to some of the specific comments.

**Specific comments**

General remark: Considering that most of the comments attached by Referee #1 in the PDF file were of an editorial/stylistic nature, with which we agree and for which we are grateful, we are not listing every line (comment) here to save time for the Editor and Referee reading the response. Therefore, if a line with a Referee's suggestion is not mentioned below, it means that we agree with the comment and will implement it in the revised version of the manuscript. Below, we only provide responses to those suggestions that, in our opinion, require clarification.

L. 77: wellfield

REPLY: We think that in British English, which we use across the manuscript, "well field" is more proper and commonly used. Thus, we suggest leaving it as is.

L. 90: Perhaps swap the order of figures 1 & 2, i.e put regional figure first ...then local

REPLY: We agree. That is a very good idea and will be implemented.

L. 415: untreated effluent?

REPLY: We think that "untreated groundwater" fits better here.

University of Silesia in Katowice
Faculty of Natural Sciences
Będzińska 60, 41-200 Sosnowiec, Poland
phone: +48 32 89 400, +48 32 2009 351, e-mail: wnp@us.edu.pl

www.**english**.us.edu.pl

[Figure]

[Figure]

HR EXCELLENCE IN RESEARCH

L. 415: This comment is hanging,,needs a context

REPLY: If we got it right, the Referee referred to the part "…(regarding raw water in the production wells)…". We agree that we have used a thought shortcut here, which may not be evident to everyone. Our point was that (at least in the case of the Kępa Bogumiłowicka RBF site) at groundwater abstraction sites using a siphon system, groundwater from individual wells is rarely tested, e.g. twice a year, because from the perspective of waterworks, the key issue is the quality of water from the collector well, particularly after the treatment process. Therefore, we wanted to specifically mention in brackets that we are referring to water from individual wells, as water from the collector well is tested much more frequently, which would contradict the point of this paragraph. Nevertheless, after rereading it, we think the sentence from the bracket can be merged with the previous sentence. Hence, we suggest:

"…Therefore, knowledge of these seasonal isotopic dynamics enables targeted, event-driven monitoring rather than merely routine observations of untreated groundwater in the individual production wells several times a year…"
* * *
University of Silesia in Katowice
Faculty of Natural Sciences
Będzińska 60, 41-200 Sosnowiec, Poland
phone: +48 32 89 400, +48 32 2009 351, e-mail: wnp@us.edu.pl

www.**english**.us.edu.pl

[Figure]

[Figure]

HR EXCELLENCE IN RESEARCH

---

## Author Comment (AC2)

 UNIVERSITY OF SILESIA
IN KATOWICE

Sosnowiec, 2 October 2025

**Authors' response to Referee #2**

**RC2: 'Comment on egusphere-2025-3841', Anonymous Referee #2, 30 Sep 2025**

**General comment**

The manuscript presents a novel application of EEMMA to RBF in Poland and reveals a strong link between isotopic evidence and catchment processes. The Materials and Methods section is comprehensive and detailed enough, providing clear descriptions of the study site, thorough field and lab methodology, transparent datasets, analytical approaches and clear methodological insights into EEMMA application. The discussion and conclusion provide an international context for the results and offer practical recommendations to RBF operators, justifying high-resolution monitoring.

The primary concerns relate to the language style, with overly long sentences and some redundant phrasing that disrupts the flow. First and foremost, however, the authors should clarify the nomenclature of water drawn from RBF wells. In my opinion, the term' groundwater abstraction' is not appropriate in the context of water drawn from the RBF wells. As you correctly pointed out in the Introduction, it is a mixture of surface water and groundwater. I hereby propose changing it just to 'water abstraction' or another term that emphasises that it is a mixture of water from different sources. In addition, it would be beneficial to briefly address the geological structure and hydrogeological conditions in the upstream catchment area, and to include a discussion how these aspects influence the results.

I would express my appreciation for the comprehensive supplementary data.

Considering the innovative character of the study and its effective presentation I recommend the manuscript for publication with minor corrections. Please refer to the attached PDF file for details.

I would like to congratulate the authors on their hard work and wish them the best of luck with the final publication.

REPLY: We would like to thank you for your time and work to review our manuscript. We are pleased you found our work innovative and suitable for HESS. We also appreciate your insightful comments provided in the PDF. We generally agree with the comments, but in a few cases, we provide an alternative explanation below.

**Specific comments**

General remark: Similar to our response to Referee #1, we did not list every comment from the attached PDF file here, since many of them were also of an editorial/stylistic nature. Please note that if a line with a Referee's suggestion is not mentioned below, it means that we agree with the comment and will implement it in the revised version of the manuscript. Therefore, below we present only responses to suggestions that, in our opinion, require more extensive commentary.

University of Silesia in Katowice
Faculty of Natural Sciences
Będzińska 60, 41-200 Sosnowiec, Poland
phone: +48 32 89 400, +48 32 2009 351, e-mail: wnp@us.edu.pl

www.**english**.us.edu.pl

[Figure]

[Figure]

HR EXCELLENCE IN RESEARCH

L. 40: I think that groundwater management is not appropriate here. Do you mean the management of water abstracted from RBF facilities?

REPLY: Yes, we will correct this part of the manuscript, but please see also our response to L. 76 regarding groundwater/water/bank filtrate terminology.

L. 66-67: Unclear what this refers to / This sentence is incomprehensible. Please rephrase it.

REPLY: This part will be corrected. We suggest:

"…Based on this example, we present recommendations for water monitoring and data analysis for RBF site operators. We believe that effective monitoring of an RBF facility requires considering data from the upstream catchment of the river and the interpretation of multiple tracers. Therefore…"

L. 76: In my opinion, the term' groundwater abstraction' is not appropriate in the context of water abstracted from the RBF system. As you correctly pointed out in the Introduction, it is a mixture of surface water and groundwater. I hereby propose changing it to 'water abstraction' or another term that emphasises that it is a mixture of water from different sources. I suggest that the corrected term be consistently used throughout the manuscript. The term' groundwater' should be reserved exclusively for groundwater.

REPLY: We fully understand and appreciate your comment and agree with it from the hydrogeological point of view. Indeed, the abstracted water may be predominantly bank filtrate or a surface water–groundwater mixture, depending on which wells we refer to.

However, since the water is drawn from the subsurface (or underground) saturated zone by production wells, we believe it is still valid to use the term *groundwater* in the paper (since, technically, it is not wrong). In our view, this terminology helps maintain clarity, especially for readers who are not specialists/experts in our field, as they intuitively associate any subsurface water extracted by wells with groundwater, regardless of the type of well recharge sources.

That being said, to address your concern, we propose a slight revision of the manuscript to clarify this point explicitly. At the beginning of Section 2.1 (Study area), we propose stating that when we refer to groundwater in the production wells, we consider that its main contribution can be linked to bank filtrate/mixture. The corresponding text can be adjusted as follows:

"…The investigated RBF site is located in the village of Kępa Bogumiłowicka near Tarnów, Poland, and is run by the public utility company Tarnów Waterworks Ltd. By groundwater abstraction, it provides about 30% of the drinking water supply for the Tarnów region. We want to emphasise that, although we use the term "groundwater" for convenience when referring to water abstracted by the production wells at the RBF site, its main component may in fact be bank filtrate or a mixture of river water and native groundwater, depending on the well location. …"

Figure 2 (it will be Figure 1, considering Referee #1's comment): The boundary between the Slovak and Polish parts of the catchment area is invisible. Consider signing the names of countries, changing line pattern or colouring the polygons.

REPLY: We will correct the Figure as suggested.

L. 96: It is worthwhile to briefly mention the geological structure and hydrogeological conditions in the catchment area upstream of the RBF system and comment in the discussion how this aspects influence results.

University of Silesia in Katowice
Faculty of Natural Sciences
Będzińska 60, 41-200 Sosnowiec, Poland
phone: +48 32 89 400, +48 32 2009 351, e-mail: wnp@us.edu.pl

www.english.us.edu.pl

[Figure]

[Figure]

HR EXCELLENCE IN RESEARCH

[Figure]

REPLY: We agree that the upstream catchment geology and hydrogeology undoubtedly shape river water characteristics. However, in our view, a detailed description of these aspects would go beyond the main scope and objectives of the paper, which is focused on tracer-based monitoring and methodological insights into RBF site management. The upstream influences are already reflected in the isotopic and hydrochemical signals analysed, and we preferred to keep the discussion concise and centred on the methodological framework – we would not want to extend it further at this point. We acknowledge, however, that a more detailed investigation of catchment geology–hydrology linkages could be an interesting direction for future research. Therefore, we cite a very comprehensive report in the manuscript that provides a holistic description of the hydrogeological conditions of the entire Dunajec River catchment:

Kruk, L., Kapuściński, J., Leśniak, J., Górka, J., Reczek, D., Biedroński, G., Hotloś, Ł., Orlak, M., Tkaczuk, W., Bubrowski, T., Augustyn, K., Czechowska, B., Herbich, P., and Przytuła, E.: Hydrogeological documentation establishing the disposable groundwater resources of the balance areas: the Dunajec River catchment area and the Czarna Orawa River catchment area. National Geological Archive, Polish Geological Institute – National Research Institute, Warsaw., 2017.

To meet expectations, we will add a sentence to the manuscript referring the reader directly to this document in the context of the hydrogeology of the entire catchment area:

"…The unconfined Pleistocene aquifer, composed mainly of alluvial gravel and sand with well-rounded pebbles, overlies thick Miocene clays and is the area's only aquifer. The RBF site is covered mainly by thin Holocene silts and organic mud; further inland, clayey silt lenses also occur (Fig. 3). The hydrogeological conditions for the entire Dunajec River catchment are described in detail in the documentation carried out by Kruk et al. (2017)…"

L. 105: I suggest using the proposed abbreviation RBF consistently throughout the text when referring to riverbank filtration.

REPLY: We will correct this.

L. 275: Following the correction of the term relating to water abstracted by RBF wells, it is likely that the use of "native" will no longer be necessary. I therefore suggest removing it from the entire manuscript.

REPLY: Please see our response to L. 76. Following this logic, we insist on staying with "native groundwater" in the paper, as we define it in L. 129 as "regional groundwater recharged via precipitation into the Tarnów aquifer system".

L. 280: Sentence correction needed for clarity. I propose: Mean daily flow rates of the Dunajec River ..... or The Dunajec River mean daily flow rates ......

REPLY: We will correct with "Mean daily flow rates of the Dunajec River".

L. 307-309: In my opinion this is a discussion paragraph.

REPLY: Even though it may be a sentence suitable for a Discussion, we propose to leave it as it is, since it directly provides the reader with information on why this Section (3.2) is needed in the manuscript, and we think it also fits well in this place.

L. 349: Please, do consider replacing the term 'river water in groundwater ...' with another term.

REPLY: Following the logic described in the reply regarding L. 76, we propose to leave the sentence as is.

University of Silesia in Katowice
Faculty of Natural Sciences
Będzińska 60, 41-200 Sosnowiec, Poland
phone: +48 32 89 400, +48 32 2009 351, e-mail: wnp@us.edu.pl

www.english.us.edu.pl

[Figure]

[Figure]

HR EXCELLENCE IN RESEARCH

L. 351-352: This sentence is unclear. Firstly, there is no connecting clause to help the reader understand the text. Secondly, could you clarify whether you are suggesting that groundwater constituted a much larger proportion in wells located further inland from the river?

I believe that the clarity of the research could be improved by applying this convention or a similar one consistently throughout the manuscript.

REPLY: In line with the comment, we suggest a slightly more detailed description:

"…River water fractions in the groundwater of the production wells, considering all used tracers, were 92–100% for S31, 87–100% for S36, 39–51% for S37, and 46–59% for S39 (mean values ± standard error of the estimated mixing fraction (SE); cf. Table S8). Wells S31 and S36 are located in a row closer to the Dunajec (Fig. 2), where a dominance of river water recharge was observed. On the other hand, results from wells S37 and S39, located in a farther row (Fig. 2), demonstrated that native groundwater inflow constituted a notably higher fraction of the groundwater abstracted from these wells…"

L. 353-354: The sentence is unclear.

REPLY: In this part, we tested how the EEMMA will work for different periods of the year, when a) isotope signals of the end-members differed, and b) isotope signals were very similar.

We propose changing the sentence to: "…The results of river water and native groundwater contribution to production wells recharge during the snowmelt period (January–June 2023) closely matched the results for the whole observation period (October 2022–October 2023; Table S8)…"

L. 366-368: This paragraph deals with the discussion.

REPLY: Our point in including this paragraph followed the same logic as the paragraph in L. 307-309: To directly inform the reader why this Section is needed in our paper and what added value it brings. Hence, we suggest leaving it as is.

L. 393: This passage is unclear. What is spatial gradient in aquifer recharge dynamics?

REPLY: The closer to the river, the more river water contributes to the water abstracted by the wells / The further from the river, the more native groundwater is abstracted by the wells.

We can add this explanation to the manuscript.

L. 394: This phrase sounds a bit awkward, please rephrase it.

REPLY: We suggest deleting this sentence from the manuscript.

L. 425: This is unclear, consider changing the wording.

REPLY: We propose: "…in the groundwater abstracted by the production wells at the RBF site…"

University of Silesia in Katowice
Faculty of Natural Sciences
Będzińska 60, 41-200 Sosnowiec, Poland
phone: +48 32 89 400, +48 32 2009 351, e-mail: wnp@us.edu.pl

www.**english**.us.edu.pl

[Figure]

[Figure]

HR EXCELLENCE IN RESEARCH

---

## Author Response (AR2)

**Authors' response to Editor**

**Editor decision: Publish as is, 06 Oct 2025**

| Public justification (visible to the public if the article is accepted and published)                                                                 |
|-------------------------------------------------------------------------------------------------------------------------------------------------------|
| Dear authors,                                                                                                                                         |
|                                                                                                                                                       |
| The revisions are adequate, so I can accept the paper for publication.                                                                                |
|                                                                                                                                                       |
| Sincerely,                                                                                                                                            |
| Gerrit de Rooij                                                                                                                                       |
|                                                                                                                                                       |
| Editor                                                                                                                                                |
| REPLY:                                                                                                                                                |
|                                                                                                                                                       |
| Dear Editor,                                                                                                                                          |
|                                                                                                                                                       |
| Once again, we would like to express our gratitude for smoothly and professionally guiding our manuscript through the review and publication process. |
|                                                                                                                                                       |
| With best regards,                                                                                                                                    |
| On behalf of my co-authors,                                                                                                                           |
| Krzysztof Janik                                                                                                                                       |
|                                                                                                                                                       |

**Authors' response to Editor**

**Editor decision: Publish subject to minor revisions (further review by editor), 02 Oct 2025**

**Public justification (visible to the public if the article is accepted and published)**

| Dear authors,                                                                                                                                                                                                                                                                                                                                                               |
|-----------------------------------------------------------------------------------------------------------------------------------------------------------------------------------------------------------------------------------------------------------------------------------------------------------------------------------------------------------------------------|
| Both reviewers were very positive about the work, and requested only minor revisions, as detailed in their respective reviews. As the reviewer recommendations never contradicted each other, I request that you implement the revisions that you propose in your reply. I do not believe another review round will be needed, so I intend to do the final check by myself. |
| Yours sincerely,                                                                                                                                                                                                                                                                                                                                                            |
| Gerrit de Rooij                                                                                                                                                                                                                                                                                                                                                             |
| Editor                                                                                                                                                                                                                                                                                                                                                                      |
|                                                                                                                                                                                                                                                                                                                                                                             |
| REPLY:                                                                                                                                                                                                                                                                                                                                                                      |
| Dear Editor,                                                                                                                                                                                                                                                                                                                                                                |
|                                                                                                                                                                                                                                                                                                                                                                             |
| Thank you very much for your message and for guiding the manuscript through the review process. We greatly appreciate the clear and constructive feedback provided, as well as your assistance at each stage.                                                                                                                                                               |
| We are especially grateful for the efficiency and timeliness of the communication throughout the process, making the experience smooth and encouraging. We carefully implemented the minor revisions as proposed in our reply to the Reviewers.                                                                                                                             |
| We look forward to the final check and are pleased with the opportunity to contribute to Hydrology and Earth System Sciences (HESS).                                                                                                                                                                                                                                        |
| With best regards,                                                                                                                                                                                                                                                                                                                                                          |
| On behalf of my co-authors,                                                                                                                                                                                                                                                                                                                                                 |
| Krzysztof Janik                                                                                                                                                                                                                                                                                                                                                             |
|                                                                                                                                                                                                                                                                                                                                                                             |

**Authors' response to Referee #1**

**RC1: 'Comment on egusphere-2025-3841', Anonymous Referee #1, 29 Sep 2025**

**General comment**

It was a pleasure to read and review this well-prepared manuscript by Janik et al.. The objectives, methods, presentation and clarity of writing are first class. The manuscript is well focussed, direct and purposeful and will be a significant contribution to RBF/MAR. It makes a valuable contribution that will be well recieved by water resources managers, especially as it gives a great deal of design and monitoring information that can be readily adapted to proposed RBF/MAR systems. I recommend that themanuscript be accepted more or less as s, subject to a few minor editorial suggestions in the attached annotated pdf file. Congratulations on an excellent paper.

REPLY: We sincerely thank you for carefully reading our manuscript and for the very positive and encouraging words. We greatly appreciate your recognition of the clarity, focus, and potential contribution of our work to the field of RBF/MAR. We are particularly grateful for your note on the practical value of the design and monitoring information, as our intention was precisely to provide material that water resources managers and practitioners can directly adapt. We also acknowledge your editorial suggestions provided in the annotated PDF. We implemented suggested revisions to improve the quality of the manuscript. Please find below our answers to some of the specific comments. Line numbers (L. ...) refer to the PDF attached by the Referee.

**Specific comments**

General remark: Considering that most of the comments attached by Referee #1 in the PDF file were of an editorial/stylistic nature, with which we agree and for which we are grateful, we are not listing every line (comment) here to save time for the Editor reading the response. Therefore, if a line with a Referee's suggestion is not mentioned below, it means that we agreed with the comment and implemented it in the revised version of the manuscript. Below, we only responded to those suggestions that, in our opinion, required clarification.

L. 77: wellfield

REPLY: We think that in British English, which we use across the manuscript, "well field" is more proper and commonly used. Thus, we left it as is.

L. 90: Perhaps swap the order of figures 1 & 2, i.e put regional figure first ...then local

REPLY: We agree. That is a very good idea, implemented in the manuscript.

L. 415: untreated effluent?

REPLY: We think that "untreated groundwater" fits better here.

L. 415: This comment is hanging, needs a context

REPLY: If we got it right, the Referee referred to the part "...(regarding raw water in the production wells)...". We agree that we have used a thought shortcut here, which may not be evident to everyone. Our point was that (at least in the case of the Kępa Bogumiłowicka RBF site) at groundwater abstraction sites using a siphon system, groundwater from individual wells is rarely tested, e.g. twice a year, because from the perspective of waterworks, the key issue is the quality of water from the collector well, particularly after the treatment process. Therefore, we wanted to specifically mention in brackets that we are referring to water from individual

wells, as water from the collector well is tested much more frequently, which would contradict the point of this paragraph. Nevertheless, after rereading it, we think the sentence from the bracket can be merged with the previous sentence. Hence, we changed this part to:

"...Therefore, knowledge of these seasonal isotopic dynamics enables targeted, event-driven monitoring rather than merely routine observations of untreated groundwater in the individual production wells several times a year..."

**Authors' response to Referee #2**

**RC2: 'Comment on egusphere-2025-3841', Anonymous Referee #2, 30 Sep 2025**

**General comment**

The manuscript presents a novel application of EEMMA to RBF in Poland and reveals a strong link between isotopic evidence and catchment processes. The Materials and Methods section is comprehensive and detailed enough, providing clear descriptions of the study site, thorough field and lab methodology, transparent datasets, analytical approaches and clear methodological insights into EEMMA application. The discussion and conclusion provide an international context for the results and offer practical recommendations to RBF operators, justifying high-resolution monitoring.

The primary concerns relate to the language style, with overly long sentences and some redundant phrasing that disrupts the flow. First and foremost, however, the authors should clarify the nomenclature of water drawn from RBF wells. In my opinion, the term' groundwater abstraction' is not appropriate in the context of water drawn from the RBF wells. As you correctly pointed out in the Introduction, it is a mixture of surface water and groundwater. I hereby propose changing it just to 'water abstraction' or another term that emphasises that it is a mixture of water from different sources. In addition, it would be beneficial to briefly address the geological structure and hydrogeological conditions in the upstream catchment area, and to include a discussion how these aspects influence the results.

I would express my appreciation for the comprehensive supplementary data.

Considering the innovative character of the study and its effective presentation I recommend the manuscript for publication with minor corrections. Please refer to the attached PDF file for details.

I would like to congratulate the authors on their hard work and wish them the best of luck with the final publication.

REPLY: We would like to thank you for your time and work to review our manuscript. We are pleased you found our work innovative and suitable for HESS. We also appreciate your insightful comments provided in the PDF attachment. We generally agreed with the comments, but in a few cases, we provided an alternative explanation below and implemented it in the paper.

**Specific comments**

General remark: Similar to our response to Referee #1, we did not list every comment from the attached PDF file here, since many of them were also of an editorial/stylistic nature. Please note that if a line with a Referee's suggestion is not mentioned below, it means that we agreed with the comment and implemented it in the revised version of the manuscript. Therefore, below we present only responses to suggestions that, in our opinion, required more extensive commentary. Line numbers (L. ...) refer to the PDF attached by the Referee.

L. 40: I think that groundwater management is not appropriate here. Do you mean the management of water abstracted from RBF facilities?

REPLY: Yes, we corrected this part of the manuscript, but please see also our response to L. 76 regarding groundwater/water/bank filtrate terminology.

L. 66-67: Unclear what this refers to / This sentence is incomprehensible. Please rephrase it.

**REPLY: Corrected with:**

"...Based on this example, we present recommendations for water monitoring and data analysis for RBF site operators. We believe that effective monitoring of an RBF facility requires considering data from the upstream catchment of the river and the interpretation of multiple tracers. Therefore..."

L. 76: In my opinion, the term' groundwater abstraction' is not appropriate in the context of water abstracted from the RBF system. As you correctly pointed out in the Introduction, it is a mixture of surface water and groundwater. I hereby propose changing it to 'water abstraction' or another term that emphasises that it is a mixture of water from different sources. I suggest that the corrected term be consistently used throughout the manuscript. The term' groundwater' should be reserved exclusively for groundwater.

REPLY: We fully understand and appreciate your comment and agree with it from the hydrogeological point of view. Indeed, the abstracted water may be predominantly bank filtrate or a surface water—groundwater mixture, depending on which wells we refer to.

However, since the water is drawn from the subsurface (or underground) saturated zone by production wells, we believe it is still valid to use the term *groundwater* in the paper (since, technically, it is not wrong). In our view, this terminology helps maintain clarity, especially for readers who are not specialists/experts in our field, as they intuitively associate any subsurface water extracted by wells with groundwater, regardless of the type of well recharge sources.

That being said, to address your concern, we propose a slight revision of the manuscript to clarify this point explicitly. At the beginning of Section 2.1 (Study area), we propose stating that when we refer to groundwater in the production wells, we consider that its main contribution can be linked to bank filtrate/mixture. The corresponding text was adjusted as follows:

"...The investigated RBF site is located in the village of Kępa Bogumiłowicka near Tarnów, Poland, and is run by the public utility company Tarnów Waterworks Ltd. By groundwater abstraction, it provides about 30% of the drinking water supply for the Tarnów region. We want to emphasise that, although we use the term "groundwater" for convenience when referring to water abstracted by the production wells at the RBF site, its main component may in fact be bank filtrate or a mixture of river water and native groundwater, depending on the well location..."

Figure 2: The boundary between the Slovak and Polish parts of the catchment area is invisible. Consider signing the names of countries, changing line pattern or colouring the polygons.

REPLY: The Figure was corrected. Also, Figure 2 is now Figure 1 based on Referee #1's suggestion.

L. 96: It is worthwhile to briefly mention the geological structure and hydrogeological conditions in the catchment area upstream of the RBF system and comment in the discussion how this aspects influence results.

REPLY: We agree that the upstream catchment geology and hydrogeology undoubtedly shape river water characteristics. However, in our view, a detailed description of these aspects would go beyond the main scope

and objectives of the paper, which is focused on tracer-based monitoring and methodological insights into RBF site management. The upstream influences are already reflected in the isotopic and hydrochemical signals analysed, and we preferred to keep the discussion concise and centred on the methodological framework – we would not want to extend it further at this point. We acknowledge, however, that a more detailed investigation of catchment geology–hydrology linkages could be an interesting direction for future research. Therefore, we cite a very comprehensive report in the manuscript that provides a holistic description of the hydrogeological conditions of the entire Dunajec River catchment:

Kruk, L., Kapuściński, J., Leśniak, J., Górka, J., Reczek, D., Biedroński, G., Hotloś, Ł., Orlak, M., Tkaczuk, W., Bubrowski, T., Augustyn, K., Czechowska, B., Herbich, P., and Przytuła, E.: Hydrogeological documentation establishing the disposable groundwater resources of the balance areas: the Dunajec River catchment area and the Czarna Orawa River catchment area. National Geological Archive, Polish Geological Institute – National Research Institute, Warsaw., 2017.

To meet expectations, however, we added a sentence to the manuscript referring the reader directly to this document in the context of the hydrogeology of the entire catchment area:

"...The unconfined Pleistocene aquifer, composed mainly of alluvial gravel and sand with well-rounded pebbles, overlies thick Miocene clays and is the area's only aquifer. The RBF site is covered mainly by thin Holocene silts and organic mud; further inland, clayey silt lenses also occur (Fig. 3). The hydrogeological conditions for the entire Dunajec River catchment are described in detail in the documentation carried out by Kruk et al. (2017)..."

L. 105: I suggest using the proposed abbreviation RBF consistently throughout the text when referring to riverbank filtration.

REPLY: Corrected.

L. 275: Following the correction of the term relating to water abstracted by RBF wells, it is likely that the use of "native" will no longer be necessary. I therefore suggest removing it from the entire manuscript.

REPLY: Please see our response to L. 76. Following this logic, we insist on staying with "native groundwater" in the paper, as we define it in L. 129 as "regional groundwater recharged via precipitation into the Tarnów aquifer system".

L. 275-276: Split the sentence / Refrase the sentence, e.g. which was the most enriched and demonstrated the least variation.

REPLY: Corrected.

L. 280: Sentence correction needed for clarity. I propose: Mean daily flow rates of the Dunajec River ..... or The Dunajec River mean daily flow rates ......

REPLY: Corrected with "Mean daily flow rates of the Dunajec River".

L. 307-309: In my opinion this is a discussion paragraph.

REPLY: Even though it may be a sentence suitable for a Discussion, we propose to leave it as it is, since it directly provides the reader with information on why this Section (3.2) is needed in the manuscript, and we think it also fits well in this place.

L. 349: Please, do consider replacing the term 'river water in groundwater ...' with another term.

HR EXCELLENCE IN RESEARCH

**REPLY: We corrected this sentence.**

L. 351-352: This sentence is unclear. Firstly, there is no connecting clause to help the reader understand the text. Secondly, could you clarify whether you are suggesting that groundwater constituted a much larger proportion in wells located further inland from the river?

I believe that the clarity of the research could be improved by applying this convention or a similar one consistently throughout the manuscript.

REPLY: In line with the comment, we extended the paragraph with a slightly more detailed description:

"...River water fractions in the groundwater of the production wells, considering all used tracers, were 92–100% for S31, 87–100% for S36, 39–51% for S37, and 46–59% for S39 (mean values ± standard error of the estimated mixing fraction (SE); cf. Table S8). Wells S31 and S36 are located in a row closer to the Dunajec (Fig. 2), where a dominance of river water recharge was observed. On the other hand, results from wells S37 and S39, located in a farther row (Fig. 2), demonstrated that native groundwater inflow constituted a notably higher fraction of the groundwater abstracted by these wells..."

L. 353-354: The sentence is unclear.

REPLY: In this part, we tested how the EEMMA will work for different periods of the year, when a) isotope signals of the end-members differed, and b) isotope signals were very similar.

We changed the sentence to: "... The results of river water and native groundwater contribution to production wells recharge during the snowmelt period (January–June 2023) closely matched the results for the whole observation period (October 2022–October 2023; Table S8)..."

L. 366-368: This paragraph deals with the discussion.

REPLY: Our point in including this paragraph followed the same logic as the paragraph in L. 307-309: To directly inform the reader why this Section is needed in our paper and what added value it brings. Hence, we suggest leaving it as is.

L. 393: This passage is unclear. What is spatial gradient in aquifer recharge dynamics?

REPLY: The closer to the river, the more river water contributes to the water abstracted by the wells / The further from the river, the more native groundwater is abstracted by the wells.

We added this short explanation to the manuscript.

L. 394: This phrase sounds a bit awkward, please rephrase it.

REPLY: We deleted this part from the manuscript.

L. 425: This is unclear, consider changing the wording.

REPLY: We changed the sentence: "...in the groundwater abstracted by the production wells at the RBF site..."